# Interactions of the Insect-Specific Palm Creek Virus with Zika and Chikungunya Viruses in *Aedes* Mosquitoes

**DOI:** 10.3390/microorganisms9081652

**Published:** 2021-08-03

**Authors:** Cassandra Koh, Annabelle Henrion-Lacritick, Lionel Frangeul, Maria-Carla Saleh

**Affiliations:** Viruses and RNA Interference Unit, Institut Pasteur, CNRS UMR 3569, 75015 Paris, France; annabelle.henrion-lacritick@pasteur.fr (A.H.-L.); lionel.frangeul@pasteur.fr (L.F.)

**Keywords:** insect-specific virus, arbovirus, mosquito, *Aedes*, ISV–arbovirus interference

## Abstract

Palm Creek virus (PCV) is an insect-specific flavivirus that can interfere with the replication of mosquito-borne flaviviruses in *Culex* mosquitoes, thereby potentially reducing disease transmission. We examined whether PCV could interfere with arbovirus replication in *Aedes* (*Ae.*) *aegypti* and *Ae. albopictus* mosquitoes, major vectors for many prominent mosquito-borne viral diseases. We infected laboratory colonies of *Ae. aegypti* and *Ae. albopictus* with PCV to evaluate infection dynamics. PCV infection was found to persist to at least 21 days post-infection and could be detected in the midguts and ovaries. We then assayed for PCV–arbovirus interference by orally challenging PCV-infected mosquitoes with Zika and chikungunya viruses. For both arboviruses, PCV infection had no effect on infection and transmission rates, indicating limited potential as a method of intervention for *Aedes*-transmitted arboviruses. We also explored the hypothesis that PCV–arbovirus interference is mediated by the small interfering RNA pathway in silico. Our findings indicate that RNA interference is unlikely to underlie the mechanism of arbovirus inhibition and emphasise the need for empirical examination of individual pairs of insect-specific viruses and arboviruses to fully understand their impact on arbovirus transmission.

## 1. Introduction

Mosquitoes are common vectors of arthropod-borne viruses (arboviruses) that are responsible for many diseases of significant public health burden. Particularly, the *Aedes aegypti* and *Ae. albopictus* mosquitoes contribute heavily to the emergence and transmission of the major arboviruses affecting human populations in tropical and subtropical regions. Studies using molecular tools or next generation sequencing on wild mosquitoes have revealed that they are ubiquitously infected with other viruses belonging to the same families as most arboviruses, namely Flaviviridae, Togaviridae, and Phenuiviridae [1,2,3,4,5,6]. Some of these viruses have been shown to be insect-specific viruses (ISVs)—unable to replicate in vertebrate cells [4,5,6]. Indeed, several investigations have reported that wild *Ae. aegypti* across several geographical locations carry a stable natural virome that consists of multiple ISVs [7,8,9]. In contrast, *Ae. albopictus* remains less studied, despite its major role as a bridge vector connecting sylvatic cycles of transmission involving wild animals to urban cycles involving humans [10].

Mosquito ISVs are fascinating on two counts. First, they are thought to be principally maintained through vertical transmission. Thus, like arboviruses, they must produce persistent infections in their hosts [11]. In insects, a persistent state of infection is achieved through fine control of the replication rate of viral pathogens by the sequence-specific RNA interference (RNAi)-based antiviral immune response. Second, they can influence the susceptibility of their hosts to arboviruses [11,12,13]. Through mechanisms that are still unconfirmed, co-infecting ISVs can interact with arboviruses, resulting in the suppression or enhancement of arbovirus replication, dissemination, and transmission rates [6,9,14,15,16,17,18]. Thus, the biology of ISV infections and their interactions with arboviruses are becoming indispensable aspects of arboviral disease ecology.

There is also special interest in utilising ISVs as an intervention to control the spread of arboviral diseases [11,19,20]. An example of this is a series of work on Palm Creek virus (PCV), an insect-specific flavivirus isolated from *Coquillettidia xanthogaster* mosquitoes in Australia [6]. In *Aedes albopictus* C6/36 cells, PCV restricts the replication of the Kunjin subtype of West Nile virus (KUNV) and Murray Valley encephalitis virus (MVEV) [6]—both members of the Japanese encephalitis serological complex of mosquito-borne flaviviruses. In *Culex annulirostris* mosquitoes, the main vector of KUNV and MVEV in Australia, the infection and transmission rates of KUNV are reduced when mosquitoes have been artificially infected with PCV before being challenged with KUNV [14]. Notably, PCV was observed to inhibit only viruses from the same genus. PCV had no effect on the replication of Ross River virus, genus *Alphavirus* [6].

Here, we sought to understand how PCV affects the vector competence of *Aedes* mosquitoes given their significant role in arbovirus circulation. We first investigated the persistence of PCV infection in *Ae. aegypti* and *Ae. albopictus* mosquitoes with a focus on heritability. We then assayed the effects of a PCV pre-infection on vector competence for Zika and chikungunya viruses, which respectively belong to the genera *Flavivirus* and *Alphavirus*. Our results have implications on the potential applications of PCV to limit the transmission of *Aedes*-transmitted arboviruses and shed light on possible ISV–arbovirus interference mechanisms.

## 2. Materials and Methods

### 2.1. Mosquitoes, Cells and Viruses

Experiments in this study used a colony of *Ae. aegypti* mosquitoes collected in Thailand (22–23 generations from the field) and a colony of *Ae. albopictus* mosquitoes collected in Vietnam (27–28 generations from the field). Mosquitoes were reared under standard insectary conditions (28 °C and 70% relative humidity under a 12/12 h light/dark cycle). Hatched larvae were reared on TetraMin fish flakes (TetraMin, Melle, Germany) at a density of 200 larvae per one litre of water until pupation. Adult mosquitoes were kept in 30 cm × 30 cm × 30 cm cages and given 10% sucrose solution to feed on ad libitum until used for experiments at four days post-eclosion. 

*Ae. albopictus*-derived C6/36 (ATCC CRL-1660) cells for the propagation of Palm Creek virus were maintained at 28 °C in Leibovitz’s L-15 medium (Gibco, Life Technologies, Paisley, UK) supplemented with 10% (by volume) of foetal bovine serum (FBS; Gibco), 1% (by volume) of penicillin-streptomycin (P-S; Gibco), 1% (by volume) non-essential amino acids (NEAA; Sigma, Irvine, UK), and 2% (by volume) tryptose phosphate broth (Sigma, Irvine, UK). Mammalian BHK-21 (ATCC CCL-10) and Vero (ATCC CRL-1586) cells were used to propagate arboviruses and for plaque assays, respectively, and were maintained at 37 °C with 5% CO_2_ in Dulbecco’s modified Eagle’s medium (DMEM; Gibco) supplemented with 10% (by volume) of FBS (Gibco) and 1% (by volume) of P-S (Gibco). 

An isolate of Palm Creek virus (PCV) was kindly provided by Roy Hall (University of Queensland, Queensland, Australia). Stocks of this virus were propagated in C6/36 cells, titrated (1 × 10^6^ TCID_50_/mL), and stored at −80 °C. Chikungunya virus (CHIKV) Caribbean strain of Asian genotype, characterised in [21], was generated from an infectious clone as described in [22]. Briefly, infectious clone RNAs were produced by in vitro transcription from a linearised plasmid using the mMESSAGE mMACHINE SP6 kit (Ambion, Life Technologies, Carlsbad, CA, USA). Purified viral RNA was transfected by electroporation into BHK-21 cells and incubated at 37 °C for 48 h. Virus stocks were then recovered from clarified cell supernatant at the end of the incubation period. After one amplification passage in BHK-21 cells, virus was titrated and stored at −80 °C. Zika virus (ZIKV) African strain MR-766 was similarly generated from an infectious clone constructed in [23]. Infectious virus was recovered from the supernatants of transfected 293T cells and passaged once in Vero cells before titration and storage at −80 °C [24].

### 2.2. Palm Creek Virus Infections in Mosquitoes

Four-day-old female mosquitoes were infected with PCV by intrathoracic injection of undiluted viral supernatant with a Nanoject III injector (Drummond Scientific Company, Broomall, PA, USA)—100 nL for infection dynamics experiments and 150 nL for arbovirus inhibition experiments. As a control for the arbovirus inhibition experiments, mosquitoes were mock-injected with spent cell culture medium.

In the infection dynamics experiments, mosquitoes were dissected at the indicated timepoints to collect midgut, ovary, and carcass tissues (*n* = 8 per timepoint). Tissues were homogenised in 100 μL of extraction buffer (10 mM Tris pH 8.2, 50 mM sodium chloride, 1 mM EDTA, with 1.25% (by volume) proteinase K (Eurobio Scientific, Les Ulis, France) with a Qiagen TissueLyser 2 (Qiagen, Hilden, Germany) for use in one-step qRT-PCRs to determine PCV titre [25]. This experiment was repeated once for ovary and carcass tissues.

### 2.3. Arbovirus Infections in Mosquitoes

Human blood for use in bloodmeal were obtained as red blood cell pellets from the Clinical Investigations and Access to BioResources platform (ICAReB) of Institut Pasteur (BB-0033-00062/ICAReB platform/Institut Pasteur, Paris/BBMRI AO203/[BIORESOURCE]). Blood was collected from healthy volunteers as per the CoSimmGen and DIAGMICOLL protocols as approved by the French Ethical Committee (DC 2008-68 COL I). 

To prepare blood for oral infection, red blood cells were washed thrice with 1× DPBS and resuspended to the original volume. Infectious bloodmeals were prepared by mixing resuspended blood (supplemented with 6% (by volume) ATP) and arbovirus dilution (supplemented with 1% (by volume) sodium bicarbonate) in 2:1 ratio to a final virus concentration of 1 × 10^6^ pfu/mL for both CHIKV and ZIKV. Prior to an infectious bloodmeal, female *Ae. aegypti* or *Ae. albopictus* adults were starved of sucrose solution overnight, counted, and placed in containers of 60 mosquitoes each. Mosquitoes were offered infectious blood for 15 min maintained at 37 °C using the Hemotek membrane feeding system (Hemotek, Blackburn, UK) through a piece of porcine intestine.

At seven days post-infection for CHIKV and at 14 days post-infection for ZIKV, mosquitoes were salivated by inserting the proboscis into 20 μL of FBS at the end of a pipette tip and allowing the mosquito to expectorate for 30 min. FBS was then diluted with 180 μL of 1× DPBS. Head, midgut, and carcass tissues were dissected out and homogenised in 200 μL of 1× DPBS with a Precellys Evolution tissue homogenizer (Bertin Instruments, Montigny-le-Bretonneux, France) for virus titration by plaque assay or RNA isolation for qRT-PCR.

### 2.4. RNA Isolation

For infection dynamics experiments, mosquito tissues homogenised in extraction buffer were readied for qRT-PCR by incubating at 56 °C for 5 min, followed by 98 °C for 5 min. For arbovirus inhibition experiments, RNA was isolated from 180 μL of homogenised mosquito in 1× DPBS with the TRIzol reagent (Invitrogen, Thermo Fisher Scientific, Waltham, MA, USA) following the manufacturer’s protocol. DNase treatment was performed on all RNA samples using a DNase I recombinant kit (Roche, Basel, Switzerland) with incubation at 37 °C for 30 min followed by 85 °C for 10 min.

### 2.5. Quantitative Reverse Transcription Polymerase Chain Reaction (qRT-PCR) of Palm Creek Virus

Titres of Palm Creek virus were quantified from RNA using the Luna Universal One-Step RT-qPCR Kit (New England BioLabs, Ipswich, MA, USA) and a primer set targeting the NS1 region of the PCV genome (Fwd 5′-TTCCCACGTGTAGTGAAGAAAGTA-3′; Rev 5′-TTTATGGCGCTGGTTAGGAC-3′). 

To create ssRNA standards for qRT-PCR, the NS1 region of PCV was reverse transcribed with Maxima H Minus Reverse Transcriptase Kit with dsDNase treatment (Thermo Scientific, Waltham, MA, USA) using a gene-specific primer (Rev 5′-TAGGTCTATTGTGTACGCGTATGG-3′) according to the manufacturer’s protocol. Template DNA was then generated from the resulting cDNA by PCR using DreamTaq DNA polymerase (Thermo Scientific) with the primer set Fwd 5′-GATGTATTGCAGTGTGCCGC-3′ and Rev 5′-TAGGTCTATTGTGTACGCGTATGG-3′. To prepare for in vitro transcription of this segment, another round of PCR was conducted, substituting the forward primer with the primer Fwd 5′-GGATCCTAATACGACTCACTATAGGGATGTATTGCAGTGTGCCGC-3′ to introduce the T7 promoter sequence (underlined). The ssRNAs were then transcribed using the MEGAscript T7 transcription kit (Invitrogen) with a 2 h incubation time and then precipitated with lithium chloride according to the manufacturer’s protocol. Transcribed RNA was quantified using the Qubit RNA HS Assay Kit on a Qubit 3 Fluorometer (Invitrogen). A 10-fold serial dilution was performed to prepare RNA standards of 10^1^ to 10^8^ ssRNA copies.

### 2.6. Plaque Assays for Arbovirus Titration

Homogenised tissue and ZIKV saliva samples were serially diluted 10-fold in serum-free DMEM, inoculated onto a monolayer of confluent Vero cells in 24-well plates, and incubated at 37 °C for 1 h. After the inoculation period, the inoculum was removed and cells were overlaid with a 1:4 mixture of 4% (by weight) agarose and DMEM supplemented with 2% FBS, 1% P-S, and 1% Antibiotic-Antimycotic (Gibco). Cells were then incubated at 37 °C for 3 days (CHIKV) or for 5 days (ZIKV). At the end of the incubation period, cells were fixed with 4% formaldehyde solution (in water) and stained with 0.1% Crystal Violet solution (in 20% ethanol and water) to visualise plaques.

Saliva samples collected from mosquitoes infected with CHIKV in the arbovirus inhibition experiments were amplified in C6/36 cells before qualitative detection on Vero cells. In this amplification step, monolayers of C6/36 cells in 96-well plates were inoculated with 100 μL of complete L-15 medium and incubated at 28 °C for 5 days. Ten microlitres of supernatants were then transferred to a monolayer of Vero cells in 96-well plates and incubated at 37 °C for 3 days. Cells were then fixed and stained as per the plaque assay protocol above. Visible cytopathic effects in Vero cells signals the presence of infectious CHIKV particles.

### 2.7. Insect-Specific Virus Titration

PCV stocks used in mosquito injections were titrated using the Median Tissue Culture Infectious Dose (TCID_50_) method. C6/36 cells were seeded onto a 96-well plate at a density of 5 × 10^4^ cells/well and allowed to adhere overnight. A 10-fold serial dilution up to 10^−11^ was made from the virus sample in serum-free L-15 medium (Gibco). To confluent C6/36 cells in a 96-well plate, eight replicates of each dilution including a no-virus control were added and incubated at 28 °C. After 5 days, cells were examined for the presence of cytopathic effects and wells were scored. Titres expressed in TCID_50_/mL were obtained using a TCID_50_ calculator provided by Marco Binder (German Cancer Research Center, Heidelberg, Germany; downloadable from http://www.molecular-virology.uni-hd.de; accessed 8 July 2020), based on the Spearman and Kärber statistical method [26].

### 2.8. Searching for Regions of Viral Genome Similarity

To search for locations on arbovirus genomes that could be targeted by PCV-derived small interfering RNAs, the PCV genome (Genbank accession number KC505248) was fragmented into 21-mers using splitter from the EMBOSS software suite [27]. PCV 21-mers were then mapped using Bowtie (version 1.2.3) [28] against the genomes of ZIKV (HQ234498) and CHIKV (LN898104.1) strains used in this study allowing for 0, 1, and 2 mismatches in high scoring pairs. Additionally, PCV 21-mers were similarly mapped against genomes of two strains of the Kunjin subtype of West Nile virus, (KUNV) isolates MRM16 (KX394396) and K68967 (KT934802), which had been used in a previous inhibition study by Hall-Mendelin et al. [14]. Virus genome coding sequences were compared with BLASTn using permissive scoring parameters (match = + 1, mismatch = −1, gap-opening = −2, gap-extend = −1) [29].

### 2.9. Statistical Analysis

All statistical analyses were performed using the software Prism 9.0.2 (161); GraphPad Software, LLC: San Diego, CA, USA, 2021. Viral titre graphs show means and standard errors. Comparisons of viral titres were done with multiple t-tests with Holm–Šidák correction. Comparisons of the proportion of infected tissues or saliva were done with Fisher’s exact test. All raw data were made available as Appendix A. 

## 3. Results

### 3.1. PCV Persistently Infects Aedes Mosquitoes

*Ae. aegypti* and *Ae. albopictus* are the most important vectors of arboviral diseases worldwide. Although PCV has been shown to be able to infect *Ae. aegypti* intrathoracically [14,30], we thought to examine its ability to infect *Ae. albopictus*. We were particularly interested to know whether PCV infection could reach the mosquito midgut and ovaries due to potential implications this may have on arbovirus interference and vertical transmission. To that end, we intrathoracically injected mosquitoes from laboratory-bred colonies of *Ae. aegypti* and *Ae. albopictus* with a standardised dose of PCV stock (1 × 10^4^ TCID_50_/mL). Carcass, midgut, and ovary tissues were dissected and collected at several timepoints up to 22 days post-infection (dpi). PCV RNA could be detected until the end of the experiment. Virus titres peaked at 10 dpi in *Ae. aegypti* (Figure 1a) and at 14 dpi in *Ae. albopictus* (Figure 1b) and remained high thereafter. This experiment was conducted twice, with midguts only assayed in the second experimental replicate.

Unlike arboviruses, which alternate between vertebrate and arthropod hosts, ISVs are thought to be transmitted vertically or through the environment [11]. Indeed, for several ISVs, vertical transmission has been shown experimentally [31,32,33]. We thus tested whether vertical transmission of PCV could occur in *Ae. aegypti* mosquitoes. A non-infectious bloodmeal was offered to PCV-injected female adults at seven dpi when PCV titres are known to be high and the infection has disseminated to the ovaries. None of the F1 larvae (*n* = 257), females (*n* = 287), or male (*n* = 340) adults tested positive for PCV by qRT-PCR, despite high PCV titres in the bodies of F0 mothers. This suggests limited ability for PCV to be vertically transmitted in *Ae. aegypti*, at least using our experimental conditions.

### 3.2. PCV Does Not Affect Vector Competence for ZIKV or CHIKV Infection

To examine the effect of PCV pre-infection on the susceptibility of *Aedes* mosquitoes to arboviruses, we challenged PCV-injected and mock-injected mosquitoes with an infectious bloodmeal of ZIKV (*Flavivirus*) or CHIKV (*Alphavirus*). Only blood-fed females were included in these experiments. ZIKV infection experiments on *Ae. albopictus* mosquitoes could not be performed due to the refractoriness of our colony. 

For both mosquito species, PCV showed no effect on arbovirus titres relative to mock-infected mosquitoes, except for a significant decrease in ZIKV titres in *Ae. aegypti* heads by 0.823 log_10_ pfu/mL (adjusted *p*-value = 0.0034) (Figure 2a,c,e). There were no significant differences in arbovirus infection, dissemination, or transmission rates, which are calculated as the proportion of infected midguts, heads, and saliva, respectively (Figure 2b,d,f). PCV infections in dual-infected mosquitoes were ascertained and found to be at expected titres (Appendix A).

### 3.3. Lack of Genome Similarity between PCV and Arboviruses

In mosquitoes, the small interfering RNA (siRNA) pathway is one of the main antiviral immune responses [34,35] and could be one among several mechanisms underlying ISV–arbovirus interference. The pathway is activated by the presence of virus-derived double-stranded RNA (dsRNAs) and results in the degradation of complementary RNA molecules [36]. Across the studies demonstrating ISV–arbovirus interference, interaction is more likely to occur when the ISV–arbovirus pair are closely related, i.e., belonging to the same genus [6,17]. We hypothesise that simultaneous infection of viruses with highly similar genomes increases the stoichiometric availability of virus-derived dsRNA substrates for the siRNA pathway to act upon. This may lead to stronger immune response against the co-infecting arboviruses.

We put this hypothesis to the test in silico by generating 21-mers of the PCV genome (10,075 fragments in total) and searching for sequence complementarity between these fragments and the genomes of CHIKV and ZIKV allowing for 0, 1, or 2 mismatches. In addition, we performed this analysis with the genomes of KUNV (isolates K68967 and MRM16), which PCV had been reported to inhibit [6,14]. We found no perfect complementary pairs across all arbovirus genomes in this analysis. With the ZIKV genome, 5 high similarity pairs (HSPs) were found with 1 mismatch and 17 HSPs with 2 mismatches between position 9019 and 10156 of the ZIKV genome. With KUNV isolate MRM16, we found 2 HSPs with 1 mismatch and 4 HSPs with 2 mismatches between position 9081 and 9084. With KUNV isolate K68967, we found HSPs only when 2 mismatches were allowed (Appendix A), between position 8937 and 9010. No sequence complementarity was found with the CHIKV genome.

The lack of correlation between genome similarity and interference is reinforced at the level of genome sequences. With ZIKV and both KUNV isolates, PCV shared only a short region ranging between 2428–2452 base pairs in length with 55–56% nucleotide similarity (Table 1). All 21-mer HSPs between PCV and the arboviruses fell within this region of similarity, which encodes the non-structural protein 5 (NS5). No HSPs were found between PCV and CHIKV under our strict scoring parameters. Interestingly, in the study by Hall-Mendelin and colleagues [14], PCV infection lowered the body titres of KUNV isolate K68967 yet increased the body titres of KUNV isolate MRM16. There is slightly more similarity between the genomes of PCV and KUNV isolate MRM16 than between PCV and KUNV isolate K68967. Further, PCV infection did not produce a marked impact on ZIKV replication in our study, despite ZIKV being more similar to PCV than the KUNV isolates are (Table 1). Genome sequence similarity is therefore not a predictor of PCV–arbovirus interactions within the same genus.

## 4. Discussion

The ability of PCV and other ISVs to limit the replication of arboviruses has raised urgent questions on whether the virome of a mosquito could impact its vector competence and stimulated interest in utilising ISVs as method of intervention against the spread of arboviral diseases. PCV has emerged as an interesting ISV due to its negative impact on the replication of mosquito-borne flaviviruses and its ability to readily infect Australian colonies of *Cx. annulirostris*, *Ae. aegypti*, and *Ae. vigilax* through intrathoracic injection [14]. Our results build upon this series of work on PCV to show that it persistently infects *Ae. albopictus* as well but does not greatly influence the vector competence of *Ae. aegypti* and *Ae. albopictus* for arboviruses. 

Although *Ae. aegypti* consistently appear to be highly susceptible to ISV infection as evidenced by its diverse natural virome [7,8,9], the few studies exploring the *Ae. albopictus* viromes show that it is comparatively less hospitable to ISVs and that its virome profiles vary more widely across geographical locations [9,37]. Thus, it is interesting that PCV can easily produce a persistent infection in this species. It suggests that the scarcity of ISVs in wild *Ae. albopictus* is due to strong infection barriers, which we had bypassed in our experimental infection using an intrathoracic injection method. 

Persistent infection and presence of PCV RNA in the ovaries did not translate into heritable infection in *Ae. aegypti*. This mirrors similar work where Culex Flavivirus artificially infected into *Cx. pipiens* does not produce infected progeny, even though transovarial transmission was demonstrated by the presence of Culex Flavivirus in all the progeny of naturally infected field-caught females [33]. PCV had only been detected in female wild-caught *Cq. xanthogaster* mosquitoes, thus the vertical transmission of this virus remains to be concretely demonstrated [4,6].

One commonality shared between ISVs and arboviruses is the need to maintain productive infection throughout the mosquito lifespan while causing minimal pathogenicity to achieve successful transmission [12]. In arthropods, the RNAi-based immune response is key to the establishment of persistent infections [22,38,39]. Based on the origin of the substrate RNAs initiating the response and the effector proteins the RNAs associate with, this response comprises three distinct pathways that each lead to the sequence-specific degradation or translational repression of sequence-complementary target RNA molecules: the siRNA pathway, the microRNA (miRNA) pathway, and the PIWI-interacting RNA (piRNA) pathway [36]. Among these, the siRNA pathway is canonically known to act in antiviral defence [40].

We explored the hypothesis that siRNAs produced by the RNAi response to the PCV genome can also target the RNA genomes of co-infecting arboviruses. No perfect complementarity could be found between PCV 21-mers and any of the arbovirus genomes we studied. However, when 1 or 2 mismatches were allowed, some HSPs within the genomes of ZIKV, KUNV (isolate MRM16), and KUNV (isolate K68967) could be found. These segregated around the region coding for non-structural protein 5 (NS5)—a viral RNA-dependent RNA polymerase highly conserved among all flaviviruses [41,42,43]. Notably, no HSPs were found within the CHIKV genome, even with 1 or 2 mismatches allowed. In addition, we found no correlation between the degree of PCV–arbovirus nucleotide similarity and the occurrence of interference. Thus, there is limited evidence to suggest the siRNA activity underpins ISV–arbovirus interference. 

There is some consideration that the other pathways of the RNAi response could participate in antiviral defence [44,45,46,47]. Most of the studies demonstrating ISV–arbovirus interference in vitro had been conducted in *Ae. albopictus* C6/36 cells [6,17,48], whose deficient siRNA activity calls into question its suitability in being used to model virus–host interactions accurately [49,50]. Instead of siRNAs (21 nucleotides long), C6/36 cells have been found to produce piRNA-like small RNAs (26–32 nucleotides long) in response to virus infection [51,52]. This has been reported for an alphavirus [51], a flavivirus [53,54], and a phlebovirus [55] pathogen. Whether virus-derived miRNAs (21–22 nucleotides long) are produced during viral infection is still a highly contested conclusion [56,57]. For PCV, Lee et al. [30] performed small RNA deep sequencing on PCV-infected *Ae. aegypti* mosquitoes and found no PCV-derived miRNAs. Nevertheless, given the lack of sequence similarity between PCV and co-infecting arbovirus, it is unlikely that ISV-derived piRNAs or miRNAs are responsible for the ISV–arbovirus interference. The next likely explanation is thus superinfection exclusion due to competition for limited host resources or precious replication machinery components among closely related viruses [20]. Alternatively, ISV-induced priming of other immune pathways such as Toll, IMD, or Jak/Stat could also limit replication of a secondary viral pathogen [39].

As an important caveat to our study, we considered whether the mosquito colonies we used were infected with other ISVs. A routine check on our colonies using total RNA sequencing revealed that the *Ae. aegypti* colony harbours two ISVs (Aedes anphevirus (99% nucleotide identity to MH430666.1) and Aedes aegypti toti-like virus (94% nucleotide identity to MN053720.1)), while no ISVs were detected in the *Ae. albopictus* colony. Nothing is known about whether these two viruses are able to cause interference effects, thus it is difficult to extrapolate their implications on our results. Although wild *Ae. aegypti* mosquitoes are infected with multiple ISVs, some laboratory-bred colonies appear to be ISV-free [9]. Future in vivo studies on ISV–arbovirus interference should be accompanied by testing of ISVs in the mosquitoes involved.

Further studies on ISVs should examine whether ISV infection alters the transcription or translation of immunity-related genes to shed light on whether immune priming plays any role in ISV–arbovirus interference. For arboviruses, persistence depends on the production of DNA forms of the virus genome, called viral DNAs. It should be explored whether ISV persistence also relies on the same cellular events.

## 5. Conclusions

Our results show that PCV infects *Ae. albopictus* and *Ae. aegypti*, which suggests a potential for horizontal transmission of this virus among mosquito species that share the same habitat. Whether events of horizontal transmission could turn into vertical transmission is yet to be determined. Our results also indicate that ISV–arbovirus interference is highly context-specific. Although PCV had shown ability to interfere with arboviruses in past studies, we did not observe marked effects of PCV infection on arbovirus replication or vector competence in *Ae. aegypti* and *Ae. albopictus*, aside from a slight decrease in ZIKV titre in *Ae. aegypti* heads. We conducted an in silico analysis to test whether the siRNA pathway could underlie interference by comparing 21-mers of the PCV genome to the genomes of ZIKV, CHIKV, and KUNV. We found no perfect sequence complementarity between PCV 21-mers and the arbovirus genomes. Further, there was limited similarity when entire genome sequences were compared. Thus, ISV–arbovirus interference is not predicated on genome sequence similarity. This emphasises a need to empirically evaluate the impact of ISVs on specific arboviruses and leaves the search for underlying mechanisms open.

## Figures and Tables

**Figure 1 microorganisms-09-01652-f001:**
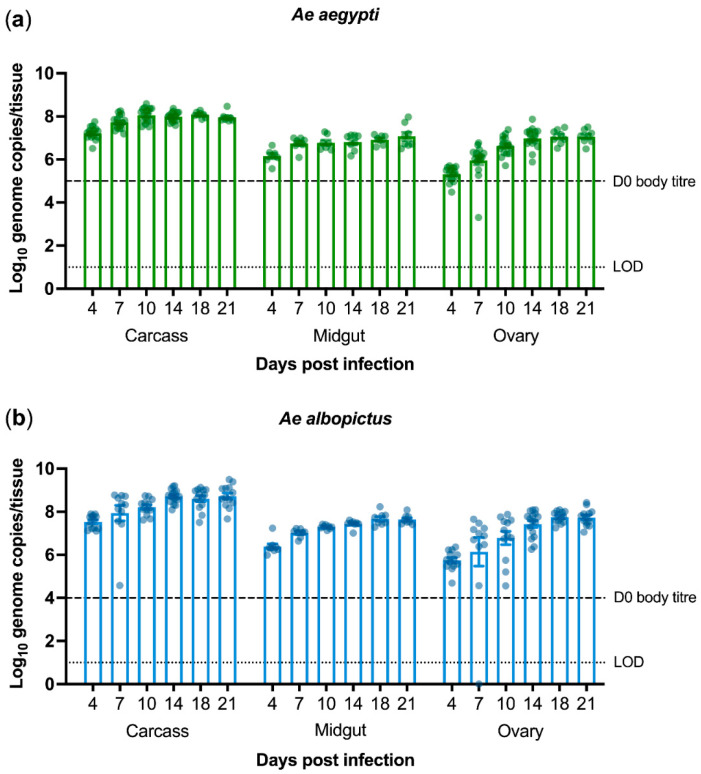
Palm Creek virus titres in intrathoracically injected *Aedes aegypti* (**a**) and *Ae. albopictus* (**b**) mosquitoes as quantified by qRT-PCR. Each data point denotes an individual mosquito from two experimental replicates. At each timepoint, carcasses and ovaries were collected from 16 individuals, whereas midguts were collected from eight individuals. Bars depicting means, standard errors, limit of detection (LOD), and virus titre of whole injected mosquitoes on the day of infection (D0) are shown.

**Figure 2 microorganisms-09-01652-f002:**
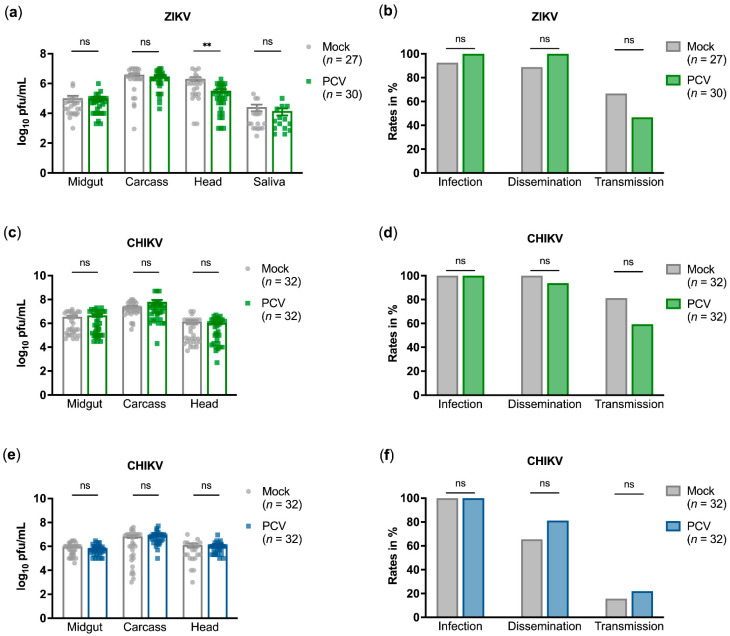
Palm Creek virus co-infection with Zika virus (ZIKV) or chikungunya virus (CHIKV) in *Aedes aegypti* (**a**–**d**) and *Ae. albopictus* (**e**,**f**) mosquitoes. (**a**,**c**,**e**) Arbovirus titres in tissues were titrated by plaque assay. Each data point denotes an individual tissue positive for arbovirus infection from two experimental replicates. Means and standard errors are shown. Mann–Whitney tests with Holm–Šidák multiple test correction were performed. (**b**,**d**,**f**) Infection, dissemination, and transmission rates as calculated by percentage of midgut, head, and saliva samples positive for infectious arbovirus, respectively. Fisher’s exact tests were performed. For all statistical tests, ns denotes no significant difference; ** denotes *p*-value < 0.01.

**Table 1 microorganisms-09-01652-t001:** Summary of BLASTn comparisons between the genomes of Palm Creek virus (PCV), West Nile virus, Kunjin subtype (KUNV isolates MRM16 and K68967), and Zika virus (ZIKV). HSP; high similarity pair. ISV–arbovirus interference is not predicated on genome sequence similarity.

PCV–Arbovirus Comparison	HSP on PCV Genome	HSP on Target Genome	Identities	Gaps	Score (bits)	*E* Value	Effect on Arbovirus
PCV–KUNV K68967	7624–9976	7819–10189	1355/2429 (55%)	134/2429 (5%)	206	6 × 10^−55^	Lower body titre [14]
PCV–KUNV MRM16	7603–9976	7872–10263	1375/2452 (56%)	138/2452 (5%)	228	2 × 10^−61^	Higher body titre [14]
PCV–ZIKV	7618–9975	7883–10249	1383/2428 (56%)	131/2428 (5%)	299	7 × 10^−83^	No effect [this study]

## Data Availability

The data presented in this study are available in Appendix A.

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
