# Peer review of "Interactions of the Insect-Specific Palm Creek Virus with Zika and Chikungunya Viruses in *Aedes* Mosquitoes"

_microorganisms, 2021, doi:10.3390/microorganisms9081652_

Round 1
Reviewer 1 Report
Insect-specific viruses have been used in some co-infection studies with mosquito vectors of disease to reduce arbovirus infection, of which can cause mammalian disease. Hence, these insect-specific viruses provide a possible biological control agent that could be used to reduce pathogenic arbovirus transmission to mammals. Basic research questions using in vitro and in vivo virus-infected mosquito systems are important to pursue. This article focuses on this work.
This was a very enjoyable research article to read. I learned a lot from the article's introduction, methods, results, and discussion.
The experiments and results addressed the researchers' questions and goals of the article.
The article is in good shape; however, some minor grammatical, word use, sentence structure revision throughout the text is necessary to make it even stronger.
Author Response
The article is in good shape; however, some minor grammatical, word use, sentence structure revision throughout the text is necessary to make it even stronger.
We thank Reviewer 1 for their kind words and are glad they enjoyed the article. The overall text has been proofread and edits have been made to improve readability as marked by tracked changes in the resubmitted manuscript document.
Reviewer 2 Report
This study by Koh et al. examines the ability of Palm Creek virus (PCV), an insect-specific flavivirus, to interfere with the infection, dissemination and transmission of Zika virus (ZIKV), a related flavivirus, and chikungunya virus, an unrelated alphavirus, in Aedes aegypti and Ae. albopictus mosquitoes. The manuscript is well written and the data presentation is of high quality.
Please find below minor comments that should be addressed before publication:
- It should be mentioned early on in the manuscript that PCV naturally infects Aedes mosquitoes, as described in the discussion section. It then remains unclear why the authors chose to inject rather than to infect by bloodmeal, which circumvents potential natural infection and dissemination barriers.
- Lines 48-58: Please give more detail on which mosquito cells and mosquitoes this paragraph refers to (C6/36?, Culex?).
- Line 160: Please present the exponents as superscript.
- Lines 205-216: Please delete as this paragraph is a direct copy of results section 3.1.
- Line 257-259: I would remove this sentence as the data is not significant. You could equally argue that the CHIKV dissemination rate in Ae. albopictus is higher in PCV-infected mosquitoes (figure 2F). It is not clear how you would explain that transmission rates are lower if there is no effect on head titres.
- Table 1: Please add references to the column "Effect on arbovirus"
I am interested to know if there is colocalization of ZIKV and PCV in mosquitoes. I appreciate that additional experiments are uncalled for and that there may not be any suitable tools to stain tissues for both viruses. If an antibody against PCV could be obtained from collaborators, maybe the authors would consider PCV-ZIKV co-infections in U4.4 or Aag2 cells to look specifically for superinfection exclusion.
Author Response
We thank Reviewer 2 for the close reading and specific comments, which improved the quality of our manuscript. Our detailed responses are listed below.
It should be mentioned early on in the manuscript that PCV naturally infects Aedes mosquitoes, as described in the discussion section.
PCV does not naturally infect Aedes mosquitoes. We believe the original sentence in line 311 led to this confusion. We have since added the underlined part to clarify this point in line 313: “PCV has emerged as an interesting ISV due to its negative impact on the replication of mosquito-borne flaviviruses and its ability to readily infect Australian colonies of Cx. annulirostris, Ae. aegypti, and Ae. vigilax through intrathoracic injection [14].”
It then remains unclear why the authors chose to inject rather than to infect by bloodmeal, which circumvents potential natural infection and dissemination barriers.
ISVs by definition would rarely be encountered by the mosquito through a bloodmeal because they are unable to replicate in vertebrates. However, it is true that a more natural route of infection could be simulated through an infectious sucrose meal. In our hands, PCV infection through this method has not yet been successful, possibly indicating a midgut barrier of infection.
Hall-Mendelin and colleagues [14] have also demonstrated that exposure of Culex annulirostris to a bloodmeal containing PCV did not produce infection. Determinants of successful natural horizontal transmission would certainly be an interesting avenue of research to pursue.
Lines 48-58: Please give more detail on which mosquito cells and mosquitoes this paragraph refers to (C6/36?, Culex?).
Details now added.
Line 160: Please present the exponents as superscript.
Done.
Lines 205-216: Please delete as this paragraph is a direct copy of results section 3.1.
This paragraph was a result of a formatting error. It has been deleted.
Line 257-259: I would remove this sentence as the data is not significant. You could equally argue that the CHIKV dissemination rate in Ae. albopictus is higher in PCV-infected mosquitoes (figure 2F). It is not clear how you would explain that transmission rates are lower if there is no effect on head titres.
Sentence removed.
Table 1: Please add references to the column "Effect on arbovirus"
References added.
I am interested to know if there is colocalization of ZIKV and PCV in mosquitoes. I appreciate that additional experiments are uncalled for and that there may not be any suitable tools to stain tissues for both viruses. If an antibody against PCV could be obtained from collaborators, maybe the authors would consider PCV-ZIKV co-infections in U4.4 or Aag2 cells to look specifically for superinfection exclusion.
This is a great point. We agree that the next logical step is testing superinfection exclusion as the possible mechanism of PCV inhibition but is out of the scope of this manuscript to do so. We thank the Reviewer for the suggested method and we certainly will contact collaborators to obtain a PCV antibody.